# Large Language Models — the Future of Fundamental Physics?

Caroline Heneka[1], Florian Nieser[2,3], Ayodele Ore[1], Tilman Plehn[1,3], and Daniel Schiller[1]

**1** Institut für Theoretische Physik, Universität Heidelberg, Germany
**2** Heidelberg Center for Digital Humanities (HCDH), Universität Heidelberg, Germany
**3** Interdisciplinary Center for Scientific Computing (IWR), Universität Heidelberg, Germany

December 12, 2025

## Abstract

**For many fundamental physics applications, transformers, as the state of the art in learning complex correlations, benefit from pretraining on quasi-out-of-domain data. The obvious question is whether we can exploit Large Language Models, requiring proper out-of-domain transfer learning. We show how the Qwen2.5 LLM can be used to analyze and generate SKA data, specifically 3D maps of the cosmological large-scale structure for a large part of the observable Universe. We combine the LLM with connector networks and show, for cosmological parameter regression and lightcone generation, that this Lightcone LLM (L3M) with Qwen2.5 weights outperforms standard initialization and compares favorably with dedicated networks of matching size.**

# 1 Introduction

The complexity and volume of experimental data in fundamental physics is increasing dramatically right now, while our lives are simultaneously transformed by modern machine learning (ML). Cutting-edge ML-methods allow us to make optimal use of this data, combining meaningful complexity, huge data volumes, fast precision simulations, and simulation-based inference into the scientific methodology of the coming decades [1–4]. Here, the fundamental paradigm shift is that *complexity is a feature, not a problem.*

To extract complex correlations, modern network architectures like transformers are extremely powerful. This is true for data acquisition, data reconstruction, first-principle simulation, and optimal inference. Initially, using existing architectures from the ML literature proved a promising path to scientific progress. Transformers with their unprecedented expressivity have brought us to the point where performance can only be improved sustainably by working toward physics-specific requirements and by using domain-specific knowledge. The prime example for physics domain knowledge are (slightly broken) symmetries, with networks built to guarantee equivariance [5–23].

For complex data representations, symmetries can be challenging to encode explicitly. An alternative approach, learning structures and symmetries inspired by foundation models, has recently gained interest in astrophysics [24–29] and particle physics [30–39]. After imposing minimal bias on the network architecture, the goal is to learn appropriate and ideally symmetry-aware data representations from generic, large datasets. The key premise is that quasi-out-of-domain data can be leveraged to scaffold a base representation for downstream finetuning on specialized data. This allows for extremely data-efficient finetuning, even across network tasks.

Thinking this pretraining strategy to the end, there remains a gap between physics research and industry in terms of the network and dataset sizes. Large Language Models (LLMs) are comprised of over 100B parameters and trained on trillions of words. On the other hand, even in particle physics applications with cheap and precise simulations, the largest open datasets used for pretraining contain around 100M jets [40–42]. For studies with the Square Kilometer Array (SKA) [43, 44], we are typically limited to tens of thousands simulated realizations of tomographic sky maps with semi-numerical codes. For fully hydrodynamical simulators we are even more limited in terms of open datasets [45, 46]. Following these considerations to the extreme leads to the question of whether LLMs can be exploited for physics [47]. Specifically, whether the extreme gap in scale between LLMs and the typical physics networks can compensate for the shift in the modality of the data. In this paper, we explore this question for the first time quantitatively and in detail. Unlike in existing particle physics and astrophysics studies, using a pretrained LLM implies a proper out-of-domain pretraining.

We begin by reviewing state-of-the-art LLMs for a physics audience in Sec. 2. Then, in Sec. 3, we outline how the LLM is adapted for numerical data. We apply Qwen2.5-0.5B [48–50] to simulations of the cosmological 21cm signal. We develop a Lightcone LLM (L3M), attaching two connector networks to the pretrained LLM. In Sec. 4 we target with L3M a 6-dimensional regression problem of astrophysical and cosmological parameters and compare the L3M performance for pretrained and randomized LLM backbones with two reference networks, one large and one with the same number of trainable parameters as the L3M connector networks. Especially the pretrained L3M fine-tuning is extremely data-efficient and it outperforms the small reference networks, showing that the LLM with out-of-domain pretraining indeed works. Finally, in Sec. 5 we go a step further and finetune the LLM backbone itself. Here, the randomized LLM backbone do not gain anything, but a pretrained and finetuned LLM outperform dedicated networks of matching size.

## 2 Large Language Models

We review the elements of state-of-the-art LLMs from a physics perspective, beginning with the data representation via tokenization in Sec. 2.1, followed by the pretraining Sec. 2.2. Then, we describe the network architecture in Sec. 2.3 and introduce finetuning methods in Sec. 2.4 and 2.5. For in-depth reviews, we recommend Refs. [51–53].

### 2.1 Tokenization

Tokenization is a crucial step in natural language processing. It introduces a representation of language by converting a string of characters $s$ into a sequence of tokens,

$$s \longleftrightarrow (t_1, \ldots, t_n) \qquad t_i \in V . \tag{1}$$

Because the vocabulary $V$ is a finite set, each token can be assigned a unique token-id. Tokens can be considered a generalization of characters. A concrete tokenizer defines the grammar for an LLM.

There exist many algorithms to create a vocabulary of lexical tokens, Byte Pair Encoding [54] being a wide-spread choice. It starts with a base vocabulary, which can tokenize all strings in the training data. This can be all characters or, alternatively, all bytes. The most frequent adjacent token pairs are then iteratively merged and added to the vocabulary as a new token. This stops once a specified vocabulary size is reached, typically of order $10^5$. Word-Piece tokenization [55, 56] also extends a base vocabulary, but instead of merging tokens by frequency, it merges them based on high mutual information between them. Once a vocabulary is created, it remains fixed and forms the latent representation of the training text. For this study, we represent physics (simulated SKA data) as non-linguistic, numeric, tokens by embedding our data with additional networks, see Sec. 3.1.

In addition, special tokens can be added or removed afterwards to indicate non-linguistic meta-information. Typically, a special token is introduced for the start, `<|im_start|>`, and the end, `<|im_end|>`, of messages, defining the chat template. It also encodes the source of a message as: (i) the system defining the broad task of the LLM, for instance a chat bot; (ii) the user whose queries prompt the LLM; and (iii) the assistant defined by the LLM's responses. The source is appended to the start token, for example as

```
<|im_start|>system
You are a wise physics AI.<|im_end|>
<|im_start|>user
What is your favorite astrophysical experiment?<|im_end|>
<|im_start|>assistant
It is the Square Kilometer Array.<|im_end|>
```

### 2.2 Autoregressive pretraining

A language generator encodes the probability of sequences of tokens, $p(t_1, \ldots, t_n)$ in a factorized, autoregressive form,

$$p(t_1, \ldots, t_n) = \prod_{i=1}^{n} p(t_i | t_1, \ldots, t_{i-1}), \tag{2}$$

LLMs are most commonly pretrained to approximate these conditionals

$$p_\theta(t_i|t_1,\ldots,t_{i-1}) \approx p(t_i|t_1,\ldots,t_{i-1})\,, \tag{3}$$

for next-token prediction [57]. Here, $\theta$ represents the network weights. The LLM is trained by minimizing the log-likelihood of a dataset, leading to a cross-entropy loss

$$\mathcal{L} = -\sum_{i=2}^{N} \Big\langle \log p_\theta(t_i|t_1,\ldots,t_{i-1}) \Big\rangle_{p_{\text{data}}(t_i|t_1,\ldots,t_{i-1})}\,. \tag{4}$$

The prediction of $t_1$ is excluded, as there is no condition. Because the vocabulary is discrete, each conditional is a categorical distribution. For particle physics, autoregressive probabilities have been introduced for phase space directions [58] and for (generated) particles [59,60].

Next-token prediction can be considered self-supervised in the sense that no explicit labeling of text in the dataset is necessary. The objective is simply to complete partial data examples. This is a difficult task in the absence of a specialized context, and extremely large datasets are required. Modern LLMs are typically pretrained on $10^{11}$ to $10^{14}$ tokens. Given that datasets of this magnitude are collected in an unsupervised manner, the data quality has to be improved through filtering or other preprocessing steps [53]. Due to the immense computational cost of pretraining an LLM, hyperparameters must be carefully chosen ahead of time [50,61].

## 2.3 Network architecture

Next-token prediction requires a network architecture that matches the conditional structure of Eq.(2),

$$f_\theta : V^n \to \text{Cat}(V)^n \qquad f_\theta(t_1,\ldots,t_n) = \begin{pmatrix} p_\theta(t|t_1) \\ \vdots \\ p_\theta(t|t_1,\ldots,t_n) \end{pmatrix} \qquad n \in \mathbb{N}\,. \tag{5}$$

First, the network $f_\theta$ has to process sequences of varying length $n$. Second, it must enforce the correct 'causal' conditioning, e.g. that $p_\theta(t|t_1)$ is independent of $t_{i>1}$ etc. Both requirements are satisfied by transformers [62]. We decompose $f_\theta$ into four parts, so a sequence of tokens $(t_1,\ldots,t_n)$ is processed by

1. an embedding layer, which maps each discrete token to a high-dimensional latent vector,

$$E : V \to \mathbb{R}^d \qquad x_i = E(t_i) \qquad \text{with} \qquad d \sim 10^4 - 10^5\,; \tag{6}$$

2. a backbone transformer which maps between sets of latent vectors,

$$g : \mathbb{R}^{n \times d} \to \mathbb{R}^{n \times d} \qquad (x_1,\ldots,x_n) \mapsto (y_1,\ldots,y_n)\,; \tag{7}$$

3. an un-embedding map which translates a latent vector into unnormalized log-probabilities,

$$E^T : \mathbb{R}^d \to \mathbb{R}^{|V|} \qquad y_i \mapsto z_i\,; \tag{8}$$

4. a normalization of the final categorical probabilities,

$$\text{Softmax} : \mathbb{R}^{|V|} \to \text{Cat}(V) \qquad \text{Softmax}(z)_i = \frac{e^{z_i}}{\sum_{j=1}^{n} e^{z_j}}\,. \tag{9}$$

We can then write the network $f_\theta$ as

$$f_\theta = \text{Softmax} \circ E^T \circ g \circ E \,, \tag{10}$$

where the softmax and (un)embedding layers act element-wise across the sequence. The embedding layers can be represented as matrices, $E \in \mathbb{R}^{|V| \times d}$, $E^T \in \mathbb{R}^{d \times |V|}$. In some LLMs, including Qwen2.5-0.5B [48–50], weights are shared between $E$ and $E^T$. Since the embedding layers act element-wise, the backbone $g$ is responsible for learning correlations among token representations. A prototypical LLM backbone architecture based on Qwen2.5 is depicted in Fig. 1. More information about Qwen2.5 and its training can be found in App. A; in the following we describe key features and concepts.

**Self Attention [62].**  This critical building block allows the backbone to handle variable-length sequences and satisfy the causal conditioning. It defines a vector representation inspired by an orthogonal basis [63], fitting the structure of Eq.(7).

We describe Grouped Query Attention [64], used in Qwen2.5. For each input vector $(x_1, \ldots, x_n) \in \mathbb{R}^{n \times d}$, $h_Q$ query, $h_{KV}$ key and $h_{KV}$ value vectors are computed via trainable affine layers,

$$
\begin{aligned}
q_i^{(j_Q)} &= W_Q^{(j_Q)} x_i + b_Q^{(j_Q)} && \in \mathbb{R}^{d_h} && (j_Q = 1 \ldots h_Q) \,, \\
k_i^{(j_{KV})} &= W_K^{(j_{KV})} x_i + b_K^{(j_{KV})} && \in \mathbb{R}^{d_h} && (j_{KV} = 1 \ldots h_{KV}) \,, \\
v_i^{(j_{KV})} &= W_V^{(j_{KV})} x_i + b_V^{(j_{KV})} && \in \mathbb{R}^{d_h} && (d_h = d/h_Q) \,,
\end{aligned}
\tag{11}
$$

implying $h_Q$ query heads and $h_{KV}$ key-value heads. Here, $h_Q$ has to be a multiple of $h_{KV}$, so the query vectors can be divided into groups of $G = h_Q/h_{KV}$ vectors. The attention matrix is

$$A_{ij}^{(j_Q)} = \frac{q_i^{(j_Q)} \cdot k_j^{(\lfloor j_Q/G \rfloor)}}{\sqrt{d_h}} \quad \in \mathbb{R}^{n \times n} \,. \tag{12}$$

The value vectors are summed for each token, weighted by attention score according to

$$a_i^{(j_Q)} = \sum_{j=1}^{n} \text{Softmax}(A_i)_j \, v_j^{(j_Q)} \,. \tag{13}$$

The resulting vectors are concatenated into

$$a_i = \left( a_i^{(1)}, \ldots, a_i^{(h_Q)} \right) \in \mathbb{R}^d \,. \tag{14}$$

An attention mask controls the dependence of $a_i$ on specific tokens. Causal conditioning corresponds to

$$A \to A + M_{\text{causal}} \qquad \text{with} \qquad (M_{\text{causal}})_{ij} = \begin{cases} -\infty & j > i \\ 0 & \text{otherwise} \end{cases} \,. \tag{15}$$

Finally, the attention output undergoes a linear map with trainable weight matrix $W_O$,

$$x_i' = W_O a_i \in \mathbb{R}^d \,. \tag{16}$$

For $h_Q = h_{KV}$ Grouped Query Attention turns into multi-head attention [62]. During inference, each token is sampled autoregressively, and the computed key-value pairs are cached for subsequent computations. Setting $h_Q > h_{KV}$ reduces the number of cached key-value pairs, speeding up inference.

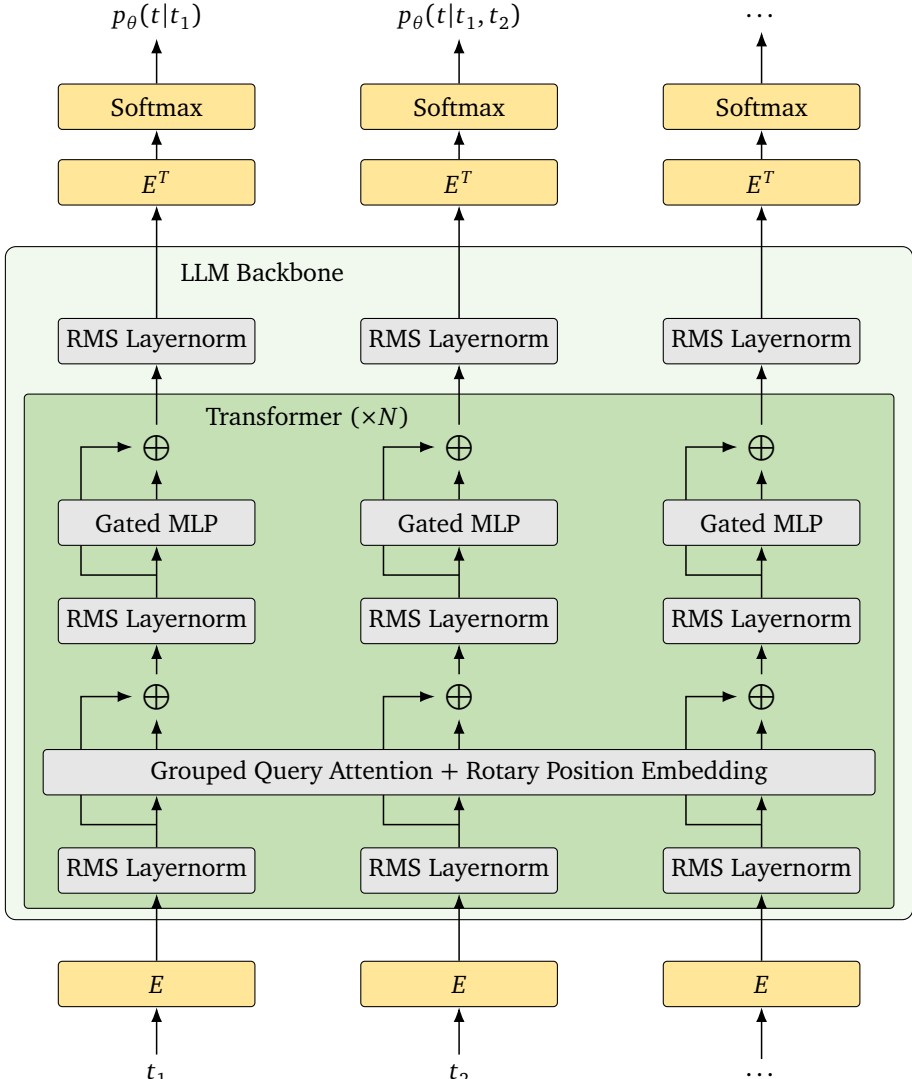

Figure 1: Qwen2.5 architecture, separating the embedding layers from the LLM backbone.

**Rotary Position Embedding [65].** The updated token representation $x_i'$ of Self Attention is manifestly invariant under permutations of the preceding token representations $(x_1, \ldots, x_{i-1})$. To add information about the relative positions between the token representations, Rotary Position Embedding is a common choice in LLMs. The scalar product in Eq.(12) is modified by inserting 2-dimensional rotations,

$$q_i \cdot_{\text{RoPE}} k_j \equiv \sum_{k=1}^{d_h/2} \begin{pmatrix} q_{i,2k} \\ q_{i,2k+1} \end{pmatrix} R((j-i)\theta_k) \begin{pmatrix} k_{j,2k} \\ k_{j,2k+1} \end{pmatrix}, \tag{17}$$

where $R((j-i)\theta_k)$ is a rotation by the angle of $(j-i)\theta_k$. The frequency $\theta_k$ depends on the dimension $k$, and is usually given by $\theta_k = \Theta^{-2k/d_h}$ with a base frequency $\Theta$. These rotations tend to give more weight to the scalar product between query-key pairs when the corresponding tokens are closer to each other.

LLMs can only reliably generate tokens if the sequence length is at most as long as the maximal trained sequence length. Since the complexity of the self-attention operation scales quadratically with the sequence length, there is a maximal trainable sequence length in prac-

tice. By freezing a pretrained LLM and training interpolating frequencies [66, 67], the supported sequence length can be extended.

**Attention dropout [68].** To reduce overfitting on dominant query-key pairs, attention dropout can be used. In this regularization technique, the entries of the softmax vectors in Eq.(13) are randomly set to zero with probability $p$, which is a hyperparameter. The non-vanishing entries of the softmax vector are scaled by a factor $1/(1-p)$.

**RMS Layernorm [69].** This operation normalizes a vector, $x \in \mathbb{R}^d$, with respect to its root mean square,

$$x'_i = \frac{\lambda_i \, x_i}{\sqrt{\frac{1}{d} \sum_{i=1}^d x_i^2 + \epsilon}} \in \mathbb{R}^d \, , \tag{18}$$

where $\lambda \in \mathbb{R}^d$ is a trainable scaling factor, which is initialized with $\lambda_i = 1$, and $\epsilon$ is a numerical cutoff. It stabilizes the training dynamics and accelerates the convergence for the deep LLMs.

**Gated MLP [70, 71].** This operation realizes a non-linear map from $\mathbb{R}^d$ to itself through a larger latent space $\mathbb{R}^{d_{\text{ff}}}$, usually with $d_{\text{ff}} = 4d$. It is defined by three trainable weight matrices, $W_1 \in \mathbb{R}^{d \times d_{\text{ff}}}$ and $W_2, W_3 \in \mathbb{R}^{d_{\text{ff}} \times d}$, and a nonlinear activation function, $\text{act}(\cdot)$,

$$x' = W_1 \left( \text{act} \left( W_2 x \right) \odot \left( W_3 x \right) \right) , \quad x, x' \in \mathbb{R}^d \tag{19}$$

where $\odot$ is the element-wise multiplication. Empirically, it outperforms standard feedforward networks in LLMs.

**Residual Connections [72].** To further stabilize the training dynamics for a deep network, residual connections are used for the Self-Attention and Gated MLP operations, indicated by $\oplus$ in Fig. 1. This structure reframes the learning objective for each block, encouraging it to learn a residual function with respect to its input rather than an entirely new representation.

## 2.4 Finetuning

In this section, we describe the typical finetuning approaches used for LLMs, which are crucial for obtaining the LLM weights.

After pretraining, the LLM has to be finetuned for a given task. A common approach is to create a dataset under supervision, which is much smaller than the one used for pretraining. The LLM is then trained further using the next-token objective of Eq.(4) on the curated dataset [73].

Reinforcement learning (RL) is another approach for finetuning an LLM [74] or to align the generated sequences with certain preferences [75]. A query sequence

$$\left( t_1^{(q)}, \dots, t_n^{(q)} \right) \equiv q \tag{20}$$

is identified as a state, and a possible response

$$\left( t_1^{(r)}, \dots, t_m^{(r)} \right) \equiv r \tag{21}$$

as a corresponding action. The LLM itself represents the conditional of the response on the query, often called the policy $\pi$

$$
\begin{aligned}
p(r|q) &= p\left(t_1^{(r)}, \ldots, t_m^{(r)} \,|\, t_1^{(q)}, \ldots, t_n^{(q)}\right) \\
&\equiv \pi\left(t_1^{(r)}, \ldots, t_m^{(r)} \,|\, t_1^{(q)}, \ldots, t_n^{(q)}\right) = \pi(r|q) \,.
\end{aligned}
\tag{22}
$$

During RL-based finetuning, a reward is assigned to each response,

$$
\text{reward}(r|q) \in \mathbb{R} \,.
\tag{23}
$$

The LLM is optimized to maximize the expected reward,

$$
\pi_{\text{optimal}} = \arg\max_\pi \left\langle \text{reward}(r|q) \right\rangle_{\pi(r|q),\, p_{\text{data}}(q)} \,.
\tag{24}
$$

Prominent RL objectives are Proximal Policy Optimization [76], Direct Preference Optimization [77] and Group Relative Policy Optimization [78].

## 2.5 Efficient training

The computational cost of finetuning can be reduced by training only a fraction of the network weights. We describe two prominent examples, which we will use in our physics study.

**Low Rank Adaptation (LoRa) [79].** Instead of training affine layers

$$
x' = Wx + b \qquad \text{with} \qquad x', b \in \mathbb{R}^{d_1}, \; x \in \mathbb{R}^{d_2}, \; W \in \mathbb{R}^{d_1 \times d_2} \,,
\tag{25}
$$

with the large matrix $W$, we can introduce a matrix $\Delta W$ as

$$
x' = (W + \alpha \Delta W)x + b \qquad \text{with} \qquad \Delta W = W_B W_A, \; W_B \in \mathbb{R}^{d_1 \times r}, \; W_A \in \mathbb{R}^{r \times d_2} \,,
\tag{26}
$$

where $W_A$ and $W_B$ are trainable, but $W$ is frozen. The combination $\Delta W$ has at most rank $r$, which is a hyperparameter. For LoRa to be effective, it must satisfy

$$
r \ll \frac{d_1 d_2}{d_1 + d_2} \,.
\tag{27}
$$

The matrix $W_B$ is typically initialized with vanishing weights, such that the weight matrix $\Delta W$ does not initially modify the output of the affine layer. For the hyperparameter $\alpha$ we choose $\alpha = 2$ throughout.

**Prompt tuning [80].** For this training technique, a new special token $x_s$ is added to the vocabulary. Then, every sequence gets prepended by this special token,

$$
(x_1, \ldots, x_n) \longrightarrow (x_s, x_1, \ldots, x_n) \,,
\tag{28}
$$

and only the embedding of this token, $E(x_s) \in \mathbb{R}^d$, is trained.

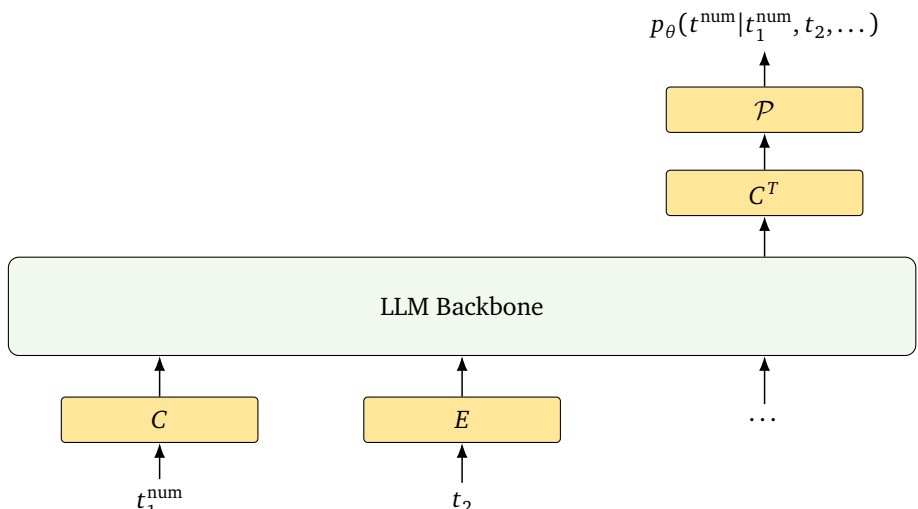

Figure 2: L3M setup connecting numerical tokens with the LLM backbone transformer.

## 3 Lightcone Large Language Model (L3M)

### 3.1 Architecture

Our goal is to see if a pretrained LLM can be used for numerical fundamental physics data and if the out-of-domain pretraining leads to a performance gain. We review a few approaches and their (dis)advantages and motivate our method:

1. We can straightforwardly express numerical data as text and query the task, as has been done for arithmetics [81, 82], regression [83, 84] and extrapolation [85]. Although LLMs can, in principle, solve these problems with in-context learning, they perform poorly and require dedicated training [86, 87]. In general, it is hugely inefficient to express numerical data as text, especially because the resulting sequences are intractably long.

2. Alternatively, we can work with multi-modal LLMs [88], which combine text and non-linguistic data. The latter is encoded with additional networks, e.g. vision transformers. The resulting embeddings are input to the LLM backbone. There are different training strategies to align the different modalities, linguistic-inspired next-token prediction is one of them. Since the generated output is text, this approach is not obviously suitable for physics.

Instead of these approaches we adapt the LLM architecture. We remember that the (un)embedding maps, $E$ and $E^T$, connect the linguistic-coded tokens with corresponding representations, for which the backbone learns correlations. We re-purpose the LLM backbone for physics data in analogy to finetuning. However, we change the modality of the pretraining and the finetuning data, so our ansatz can be viewed as model reprogramming [89]. The non-local and long-range correlations of the linguistic modality make this approach very interesting, as learning them requires a lot of computing resources.

To utilize the transformer architecture, the physics data has to be represented as a sequence of numerical 'tokens' in analogy to Eq.(1),

$$\left(t_1^{\text{num}}, \ldots, t_n^{\text{num}}\right), \qquad t_i^{\text{num}} \in \mathbb{R}^{d_{\text{num}}}. \tag{29}$$

In principle, the numerical tokens can be discrete, but they will not be in our architecture. To connect the numerical tokens to the backbone transformer we introduce input and output

connectors, $C$ and $C^T$, in analogy to the (un)embedding maps. The input connector simply maps the numerical tokens to the latent space of the backbone, while the output connector is combined with a predefined map $\mathcal{P}$ that yields a parametrization of the conditional probability $p(t_i^{\mathrm{num}}|\cdot)$. For example, in the case of linguistic tokens we have $\mathcal{P} = \mathrm{Softmax}$ and its normalized outputs define a categorical distribution. For several numerical modalities each of them gets an input and output connector network.

LLMs finetuned for time series forecasting [90–93] serve as a toy model for generative physics tasks or extrapolation. In particular, Ref. [90] re-programs the LLM backbone and achieves competitive results, supporting our L3M ansatz.

Our architecture is illustrated in Fig. 2. The input sequence starts with a numerical token, $t_1^{\mathrm{num}}$, followed by a linguistic-coded token, $t_2$. The former is connected to the LLM backbone with an input connector, $C$, and the latter with the embedding map $E$. The output connector, $C^T$, yields a parameterization of $p(t^{\mathrm{num}}|t_1^{\mathrm{num}}, t_2, \dots)$, which gets translated into a probability density by $\mathcal{P}$. For this paper, we use the small Qwen2.5-0.5B-Instruct LLM, because its limited size allows us to test different setups. The numerical tokens will be introduced in section 4 and 5.

## 3.2 21cm lightcone data

We use complex data of 21cm background fluctuations as a testbed for an LLM performing standard cosmological tasks. The Square-Kilometer-Array (SKA) [reference], as the current state-of-the-art interferometer, enables the 3D mapping of neutral hydrogen, the most abundant baryonic element in the Universe, for over 50% of the observable Universe. The 3D lightcones of the 21cm signal, 2D spatial + 1D temporal, represent the brightness temperature offset $\delta T_{21}(x, \nu)$ measured against the Cosmic Microwave Background (CMB), with on-sky coordinates $x$ and frequency $\nu$ (or equivalently, redshift $z$), as measured by a radio interferometer such as the SKA. For the regression and generative tasks in Secs. 4 and 5 we create a training dataset of several thousand lightcones.

21cm lightcones are created with the publicly available semi-numerical (approximate hydrodynamical) code `21cmFASTv3` [94, 95]. It generates initial density and velocity fields and evolves them in time, or redshift, at second-order Lagrangian perturbation theory using the Zel'dovich approximation [96]. Ionized regions are identified in an excursion set formalism by filtering the matter density field with a top-hat filter of decreasing size. A region at a certain filter scale is flagged as ionized, with a neutral fraction $x_{\mathrm{HI}} = 0$, if the fraction of collapsed matter, $f_{\mathrm{coll}}$, exceeds the inverse ionizing efficiency of star formation, $\zeta^{-1}$. Partially ionized regions are accounted for with an ionized fraction $1 - x_{\mathrm{HI}} = f_{\mathrm{coll}}\zeta$.

The resulting 21cm brightness temperature field $\delta T_{21}$ depends on ionized fraction $x_{\mathrm{HI}}$, baryonic matter density as a tracer of the underlying dark matter field, and a flat background cosmology with a cosmological constant as

$$\delta T_{21}(x, z) \approx 27\, x_{\mathrm{HI}} (1 + \delta_{\mathrm{b}}) \left( \frac{H(z)}{\mathrm{d}v_{\parallel}/\mathrm{d}r_{\parallel} + H(z)} \right) \left( \frac{1 + z}{10} \right) \left( \frac{0.15}{\Omega_{\mathrm{m}} h^2} \right)^{1/2} \left( \frac{\Omega_{\mathrm{b}} h^2}{0.023} \right) [\mathrm{mK}] \quad (30)$$

with baryonic matter fluctuations $\delta_{\mathrm{b}}(x, z)$, peculiar velocity field $\mathrm{d}v_{\parallel}/\mathrm{d}r_{\parallel}(x, z)$, Hubble function $H(z)$ for cosmological background expansion, and the matter density parameter $\Omega_{\mathrm{m}}$, Hubble parameter $h$, and baryonic matter density parameter $\Omega_{\mathrm{b}}$ at present time. In this formula we assumed the so-called post-heating regime, where the spin temperature of neutral hydrogen is significantly larger than the CMB temperature, i.e., $T_{\mathrm{S}} \gg T_{\gamma}$.

The resulting 21cm brightness offset fluctuation fields depend on several cosmological and astrophysical parameters. For our proof-of-concept study we combine parameters for cosmol-

ogy and dark matter properties, with parameters describing astrophysics during cosmic dawn and the EoR (see also [10]):

1. Matter density $\Omega_{\rm m} \in [0.2, 0.4]$
   It controls structure formation, where the chosen values encompass the Planck limits [97];

2. Warm dark matter mass $m_{\rm WDM} \in [0.3, 10]$ keV
   The prior range allows for a variety of phenomenological behavior; here the lower limit significantly deviates from a with Cold Dark Matter (CDM) scenario. Current astrophysical constraints favor mass values larger than a few keV [98, 99]. The larger $m_{\rm WDM}$, the more structure formation and the distribution of DM halos look similar to CDM, as the free-streaming length is inversely proportional to the WDM mass;

3. Minimum virial temperature $T_{\rm vir} \in [10^4, 10^{5.3}]$ K
   This parameter defines the minimum virial temperature of dark matter halos required for cooling that is efficient enough for star formation to take place. It is defined by atomic cooling limits and observations of Lyman-break galaxies [100];

4. Ionization efficiency $\zeta \in [10, 250]$
   The ionization efficiency determines if a region is flagged as ionized. It is a composite parameter determined by both star formation parameters and recombinations in the IGM via

$$\zeta = 30 \frac{f_{\rm esc}}{0.3} \frac{f_\star}{0.05} \frac{N_{\gamma/b}}{4000} \frac{2}{1 + n_{\rm rec}} \, , \qquad (31)$$

   where $f_{\rm esc}$ is the escape fraction of ionizing UV photons into the IGM, $f_\star$ is the fraction of baryonic gas bound in stars, $N_{\gamma/b}$ is the number of ionizing photons emitted per baryon by stars, and $n_{\rm rec}$ is the number density of hydrogen recombinations in the IGM, calculated for example based on local gas densities;

5. Specific X-ray luminosity $L_{\rm X} \in [10^{38}, 10^{42}]$ erg s$^{-1}$ M$_\odot^{-1}$ yr
   Integrated luminosity at energies $< 2$ keV per unit star formation rate in M$_\odot$ yr$^{-1}$ that escapes host galaxies;

6. X-ray energy threshold $E_0 \in [100, 1500]$ eV
   Energy threshold below which X-rays are absorbed by their respective host galaxies; X-rays with energies below $E_0$ do not escape the host galaxy and therefore do not contribute to heating and reionization.

Other cosmological parameters are fixed to the Planck $\Lambda$CDM values [101] and assume flatness. We take $\Omega_{\rm b} = 0.04897$, $\sigma_8 = 0.8102$, $h = 0.6766$, and $n_{\rm s} = 0.9665$.

To generate our training dataset of 21cm lightcones, we sample parameters from the uniform priors summarized in Tab. 1. For each parameter set we generate the corresponding lightcone in the redshift range $z = 5 - 35$. Each lightcone has a spatial box size of 200 Mpc at a resolution of 1.42 Mpc and consists of $(140, 140, 2350)$ voxels for 2350 temporal (redshift

| Parameter | Prior Range |
|---|---|
| Matter density $\Omega_{\rm m}$ | $\mathcal{U}[0.2, 0.4]$ |
| Warm dark matter mass in keV $m_{\rm WDM}$ | $\mathcal{U}[0.3, 10]$ |
| Minimum virial temperature in K $T_{\rm vir}$ | $\mathcal{U}[10^4, 10^{5.3}]$ |
| Ionizing efficiency $\zeta$ | $\mathcal{U}[10, 250]$ |
| X-ray energy threshold for self-absorption in eV $E_0$ | $\mathcal{U}[100, 1500]$ |
| Specific X-ray luminosity in erg/s $\log L_{\rm X}$ | $\mathcal{U}[38, 42]$ |

Table 1: Summary of the cosmological (dark matter) and astrophysical parameters sampled to simulate the 21cm signal along with their prior ranges.

or frequency) bins. We note that the matter density $\Omega_{\mathrm{m}}$ impacts the physical length in the temporal direction, as it changes the background time evolution of space-time. We therefore cut the highest-redshift voxels for a fixed number of 2350 temporal bins. Therefore, only for $\Omega_{\mathrm{m}} = 0.4$ the lightcones include $z = 35$, while smaller $\Omega_{\mathrm{m}}$ values lead to lightcones slightly cropped at high redshift (lowest frequencies).

We use our dataset of around 5000 lightcones for training, validation, and testing. We filter extreme reionization histories that are strongly disfavored by current observational bounds, in terms of the optical depth [97] and the endpoint of reionization (small fraction of neutral hydrogen) being reached at $z \sim 5$ at the latest, as indicated by measurements of the Lyman-alpha forest [102, 103]. Visualizations of the data can be found in section 4 and 5.

# 4  Parameter regression with frozen backbone

First, we examine the extent to which pretrained correlations in the LLM backbone can be utilized for physics tasks. As a benchmark task, we use the regression of simulation parameters from the 21cm lightcones, both astrophysical and related to dark matter (see Sec. 3.2 for a description of parameters and lightcone generation). To isolate the influence of pretraining, we completely freeze the backbone transformer, training only the connectors, and compare against a network where the weights of the backbone transformer are re-initialized. Any difference between the two networks can then be attributed to the pretrained LLM structure.

## 4.1  Data and connector architecture

For this regression task, we reduce the lightcones by spatially averaging the brightness temperature field, yielding the so-called global brightness temperature signal as a function of time, or redshift. In addition, we downsample the global signal by replacing 50 consecutive data points with their mean value, resulting in 47 brightness temperature values per lightcone, see Fig. 3. Each of these values is identified as a token. As preprocessing, we normalize the global signal to zero mean and unit variance and min-max normalize the 6 simulation parameters $p_i$ from Sec. 3.2 as

$$p'_i \equiv \frac{p_i - p_{i,\min}}{p_{i,\max} - p_{i,\min}} \in [0, 1] \,, \tag{32}$$

with $p_{i,\min}$ and $p_{i,\max}$ being the minimal and maximal values. The training, validation and test datasets consist of 3800, 960 and 250 lightcones, respectively.

**Architecture**   The networks follow the L3M architecture from Sec. 3.1. For regression, there are two numerical modalities: the global brightness temperature signal, $(t_1^{\mathrm{BT}}, \ldots, t_{47}^{\mathrm{BT}})$ as input and the target parameters $\vec{p}$ as output. For each of them we introduce a connector network. Large connectors improve the alignment of the numerical modalities with the linguistic token representations. On the other hand, they also reduce the importance of the backbone LLM — the connector networks may perform the regression while the backbone trivially transports the information. Since our focus is the backbone network, we use single affine layers for each connector.

We also introduce a learnable token, `<|ska-param|>`, which is appended to the input sequence after the brightness temperature tokens. The backbone embedding of this token,

$$z \equiv g\left(\texttt{<|ska-param|>} \,\big|\, t_1^{\mathrm{BT}}, \ldots, t_{47}^{\mathrm{BT}}, \ldots\right) \,, \tag{33}$$

can be interpreted as a summary embedding of the global signal, from which the simulation parameters are regressed. The final ellipsis in the above equation refers to additional tokens which we specify momentarily.

We model the systematic uncertainty of the regression as a Gaussian with a learned covariance matrix. The summary embedding $z$ is inserted into the output connector, which predicts the mean values, $\vec{\mu}$, and the covariance matrix, $\Sigma$, of the Gaussian. Consequently, the network is trained with the negative log-likelihood loss, in this context called the heteroskedastic loss [104]

$$\mathcal{L} = \frac{1}{2} \left\langle (\vec{p} - \vec{\mu})^T \Sigma^{-1} (\vec{p} - \vec{\mu}) - \log \det \Sigma^{-1} \right\rangle_{p_{\text{data}}(\vec{p} \,|\, t^{\text{BT}})}. \tag{34}$$

Due to the normalization of the parameter values from Eq. (32), the predicted mean values are activated with a sigmoid function, yielding

$$\vec{\mu} \in [0, 1]^6. \tag{35}$$

To parameterize the symmetric, positive definite covariance matrix, we employ the Cholesky decomposition, which factorizes the covariance matrix into a product of a lower triangular matrix with positive diagonal entries and its transpose,

$$\Sigma^{-1} = LL^T \qquad \text{with} \qquad L \in \mathbb{R}^{15} \times \mathbb{R}^6_+. \tag{36}$$

A softplus activation function ensures that the diagonal elements are positive. Furthermore, we divide the values in the $n$-th row of $L$ by $1/\sqrt{n}$ to unbias the initial covariance matrix. As an example, observe that

$$\begin{pmatrix} a & 0 \\ b & c \end{pmatrix} \begin{pmatrix} a & b \\ 0 & c \end{pmatrix} = \begin{pmatrix} a^2 & ab \\ ab & b^2 + c^2 \end{pmatrix}. \tag{37}$$

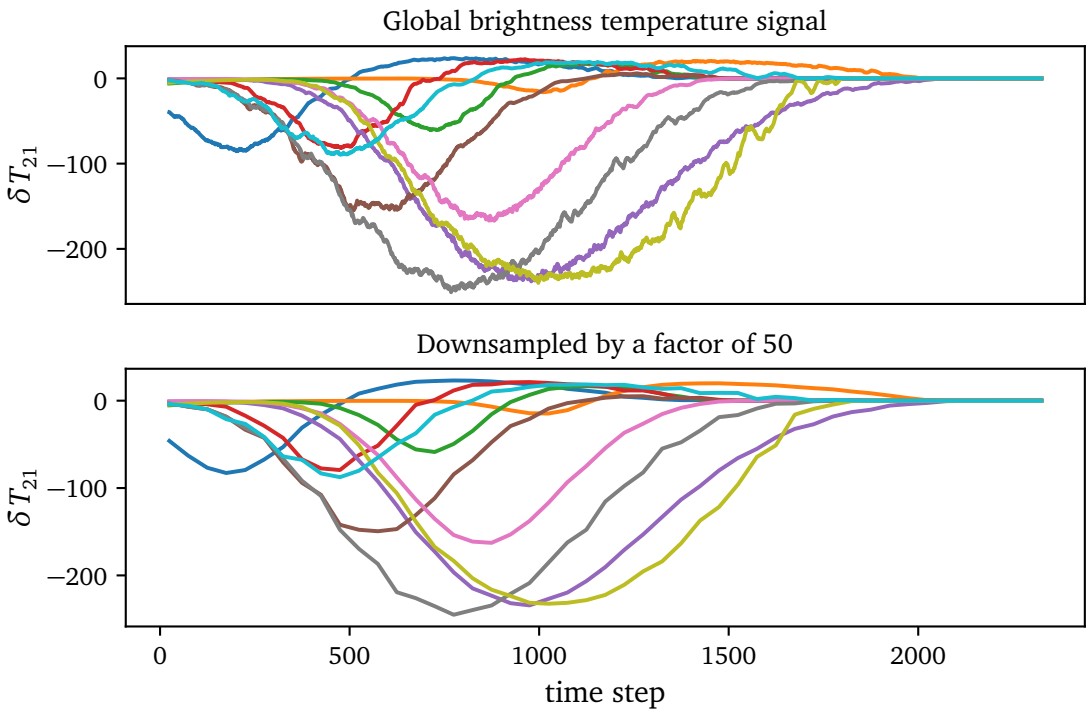

Figure 3: Global brightness temperature signal for 10 different lightcones and their corresponding downsampled distributions.

At the start of training, the network is not trained and its output can be viewed as independent random values. This implies that the above covariance matrix would have larger entries in the lower right component than in the upper left component on average.

We investigate 3 different prompting templates, containing the same information for the regression task but potentially additional (trainable) tokens. The following prompts are a visual representation of the input sequence of tokens.

1. **Minimal**  contains only the necessary tokens,

   ```
   t₁ᴮᵀ … t₄₇ᴮᵀ<|ska-param|>
   ```
   $t_1^{\text{BT}} \dots t_{47}^{\text{BT}}$`<|ska-param|>`

2. **Chat-inspired**  interleaves the numerical tokens with the chat template,

   ```
   <|im_start|>system
   <|im_end|>
   <|im_start|>user
   t₁ᴮᵀ … t₄₇ᴮᵀ<|im_end|>
   <|im_start|>assistant
   <|ska-param|><|im_end|>
   ```

   For the pretrained LLM, the tokens `<|im_start|>`, `<|im_end|>`, 'system', 'user' and 'assistant' have a pretrained embedding. For the randomly initialized network, we also randomly initialize the embeddings of these tokens and keep them frozen during training.

3. **Chat-inspired with trainable tokens**  adds, in the spirit of prompt tuning [80], two tokens `<|system-prompt-token|>` and `<|lightcone-token|>` with trainable embeddings,

   ```
   <|im_start|>system
   <|system-prompt-token|><|im_end|>
   <|im_start|>user
   <|lightcone-token|>t₁ᴮᵀ … t₄₇ᴮᵀ<|im_end|>
   <|im_start|>assistant
   <|ska-param|><|im_end|>
   ```

We remind the reader that the linguistic tokens are embedded via the pretrained embedding maps, $E$, and the numerical tokens are embedded via the added connector networks, $C$, see Fig. 2. The two connector networks have a total of 26.9k trainable parameters. The two trainable tokens introduced with the last prompt template increase this number by 1.8k.

**Training and reference networks**  The training hyperparameters are listed in the left panel of Tab. 2. We find that updating the input and output connector weights separately leads to a stable optimization. For each batch, the weights of the input connector are optimized first, after which the gradients are recomputed and the weights of the output connector are optimized. Since the dataset is relatively small, we use attention dropout to reduce overfitting. In addition, we duplicate the training dataset. This way, a batch can contain two identical samples, which become effectively distinct due to attention dropout. We empirically find that using the pretrained scaling factors of the final layer norm degrades the performance in early stages of training, so we always re-initialize those factors to ones.

To provide a reference for the regression results, we introduce two networks with the same structure as Qwen2.5, but trained from scratch:

1. a small reference network with 32k parameters
2. a larger reference network with 1M parameters

The small reference illustrates the performance of a network containing a comparable number of trainable parameters as the L3M setup, while the large reference demonstrates the expected performance of an unrestricted and dedicated network. These reference networks do not have a causal attention mask, which gives them the advantage that every token attends to every other token. We use the minimal prompt template since networks trained from scratch do not benefit from the chat template.

The network hyperparameters of the small network are determined via a hyperparameter search, while those of the large network are reasonably chosen as we do not care about its best possible performance. The choices for each network are listed in the right panel of Tab. 2. Both reference networks are trained with the setup described above, but with two adjustments: (i) they are trained for 40,000 epochs without changing the warm-up and decay periods; (ii) since all network parameters are trained, we do not use the interleaved training and update all parameters simultaneously.

## 4.2 Results

First, we look at the training efficiency of the pretrained L3M compared to the randomly initialized backbone. We show the validation loss during training in Fig. 4. The top row is grouped by prompt template: minimalist, chat-inspired, or chat-inspired with trainable tokens; the bottom row by backbone initialization: pretrained or random. In each panel, we provide just the best validation losses of the reference networks which have been trained for many more epochs. Throughout, the final validation losses of the pretrained and the randomly initialized L3Ms lie between the two reference losses, and the pretrained backbone always outperforms the random backbone. This indicates that the L3M performance is sensible.

A nontrivial common feature is that the chat template significantly boosts the performance of the pretrained network, despite adding no information to the regression task. It increases the speed of convergence and also improves the loss at each epoch. Since the embeddings for the chat template tokens are manifestly aligned with the LLM latent space through pretraining, we rationalize their benefit as providing structural information for the to-be-trained embeddings of the numerical tokens. For the randomly initialized network, the chat template has little effect, improving the network performance only at the end of training. Adding trainable tokens does not significantly affect the loss. Figure 4 shows that the pretrained LLM backbone is more computationally efficient than the random backbone.

A second observation is that even randomly initialized L3M backbone weights outperform

| | |
|---|---|
| Batch size | 1024 |
| Epochs | 1500 |
| Learning rate | $5 \cdot 10^{-5}$ |
| Learning rate schedule | 100 epochs linear warm-up, 1300 epochs stable, 100 epochs cosine decay |
| Max. gradient norm | 30 |
| Attention dropout | $10^{-3}$ |
| Optimizer | Adam |

| | | |
|---|---|---|
| Hidden dim | 128 | 32 |
| Transformer blocks | 6 | 3 |
| Query heads | 4 | 2 |
| Key-Value heads | 4 | 2 |
| MLP hidden dim | 256 | 64 |
| Number of parameters | 990k | 32K |

Table 2: Training (left) and reference network (right) hyperparameters. The number of trainable parameters of the small reference network matches the L3M connectors.

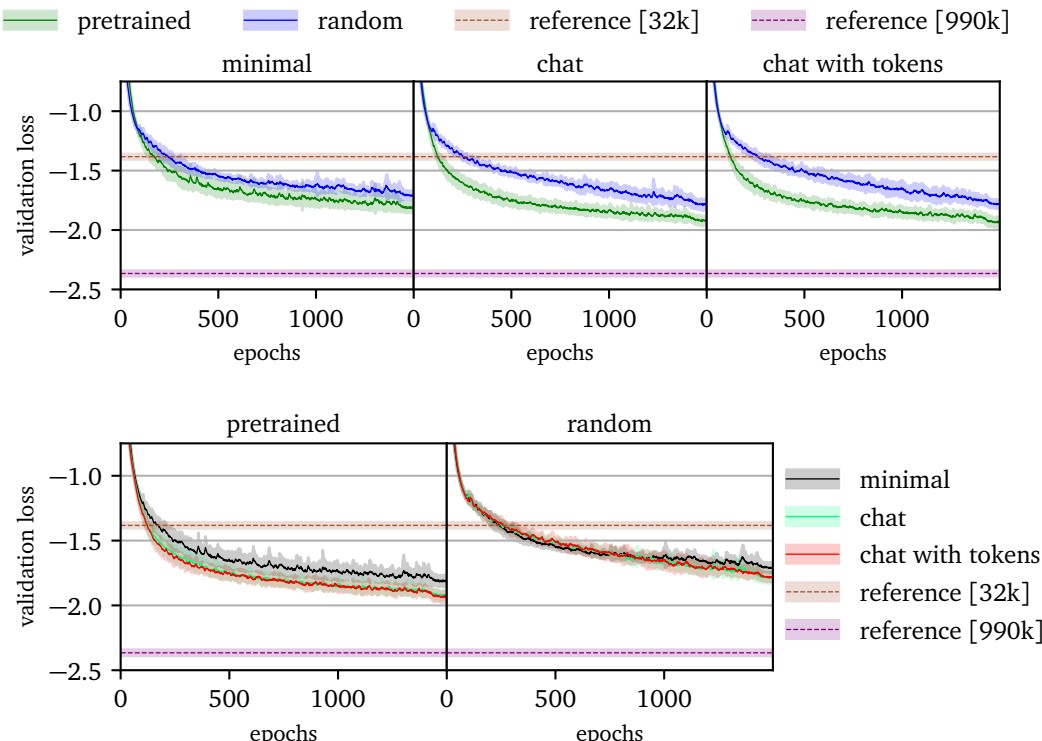

Figure 4: Mean validation loss with a $1\sigma$-band determined from 8 runs, grouped by prompting template (upper) and backbone initialization (lower). For the reference networks, only the best validation loss is shown.

the reference network with a similar number of trainable parameters. To understand this, we train the L3M with randomly initialized backbone and the minimal chat template for varying numbers of transformer blocks in the backbone. The resulting validation losses are shown in Fig. 5. Although the backbone is not trained, increasing the depth improves the network performance. We hypothesize that the deeper backbones are able to scramble the inputted information in a more expressive way. This makes it easier for the input and output connectors to access the relevant correlations.

Finally, in Fig. 6, we show the best regression results, measured by validation loss, for the astrophysical and cosmological parameters introduced in Sec. 3.2. We show results for the pretrained and randomly initialized L3Ms, as well as both reference networks. For each light-cone of the test set, the network predicts a multivariate Gaussian distribution of the underlying simulation parameters. We sample 50 times from this distribution. Since the parameter values are bounded from above and below, we clamp the sampled parameters into this interval. After that, we bin all samples with respect to the true value along the marginals of every parameter and compute the standard deviation for each bin. First, we observe that the shapes of the regressed parameters agree for all setups, including the known limitations discussed in detail in Refs. [22, 29]. Especially in the lower panels we also see that the reference network with the small number network parameters, comparable to the L3M connectors, performs much worse than both L3M setups. The performance of the pretrained L3M tends to be slightly better than the random-initialized L3M, almost matching the performance of the large reference network for many parameter values. The gap of the loss between the pretrained L3M and the large reference network seen in Fig. 4 is attributed to the latter having a smaller spread, especially for small $m_{\mathrm{WDM}}$ values and for all $\Omega_m$ values.

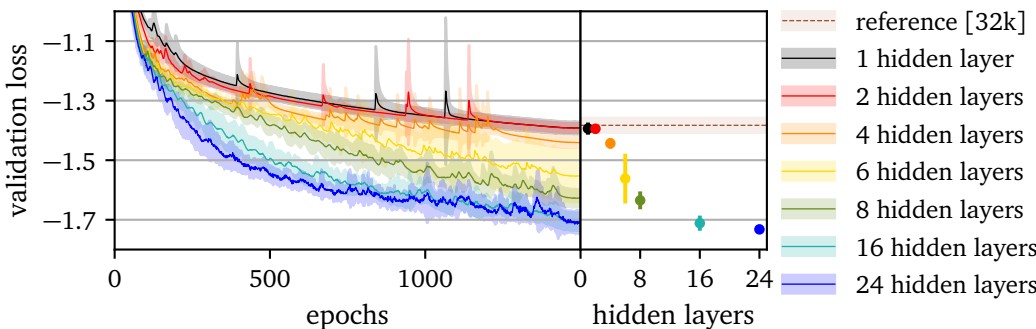

Figure 5: Mean validation loss with a $1\sigma$-band for randomly initialized backbone weights and the minimalist chat template. The number of hidden layers of the backbone network is varied. The right panel shows the best validation losses, including the small reference network.

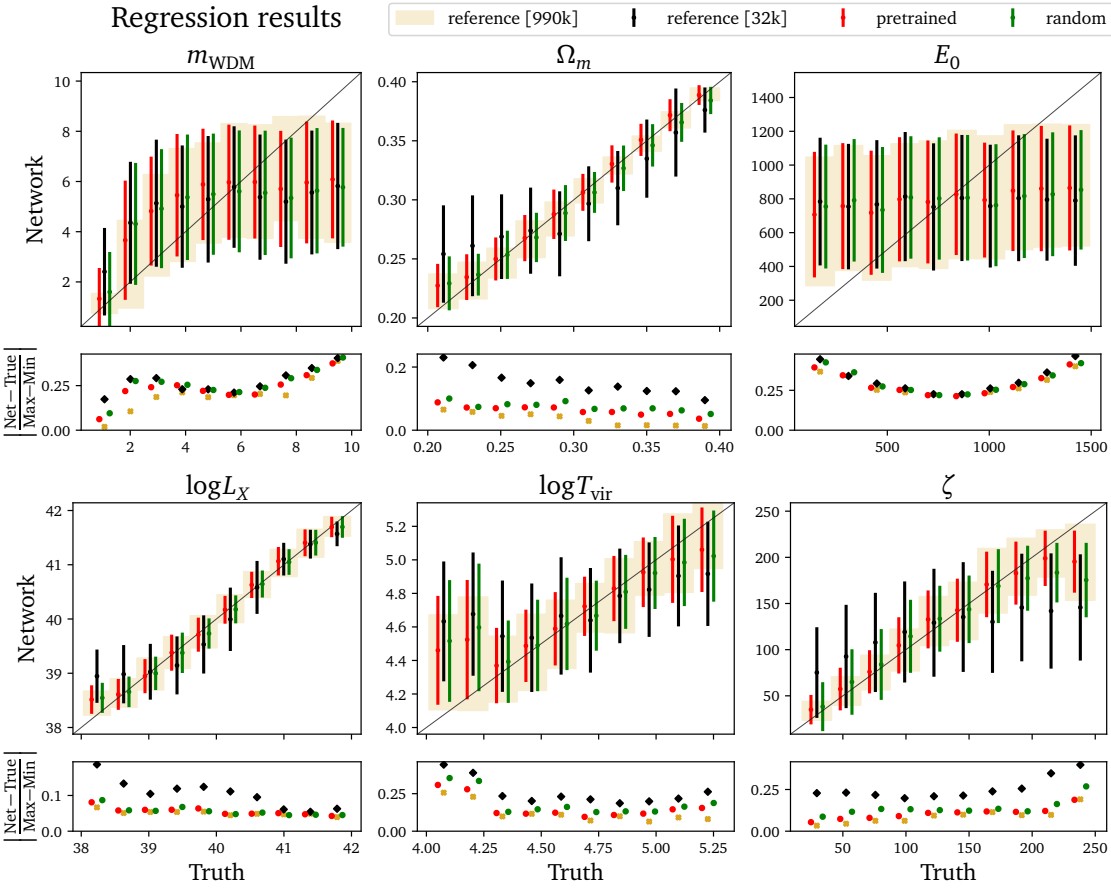

Figure 6: Regressed parameters after sampling 50 times from each regressed Gaussian distribution as described in the text.

# 5  Generation with finetuned backbone

Now, we turn to the more sophisticated task of generating slices of lightcones. For this task we will also finetune the LLM, rather than just training the connector networks on a pretrained LLM backbone.

## 5.1  Data and connector architecture

We interpret a lightcone as a time series of spatial slices, $\delta T_{BT}(\vec{x}; t)$, represented as a sequence of continuous tokens. Every spatial slice is divided into patches of $14 \times 14$ pixels and each patch is identified as a token. Then, the resulting 2d grid of patches is flattened into a sequence of patches, inserting a new-line token, `<|nl_1|>`, after every line break. Finally, the sequences of spatial slices are concatenated by interleaving another new-line token, `<|nl_2|>`, as illustrated in Fig. 7. This representation partially breaks the inherent local 3D structure of the lightcone, forcing the L3M network to recognize long-range correlations, for which the pretrained LLM backbones should be advantageous.

The generative task of the network is next-patch prediction, or next-token prediction, as discussed in Sec. 2.2. To render the sequence length feasible, we restrict to 12 consecutive spatial slices out of the 2350, which we call a sublightcone. These contain a manageable $1,200$ patches in total. The first two slices are excluded from the next-patch prediction, ensuring that each forecast slice depends on at least two preceding slices. This guarantees that a velocity field can be extracted from the context. Since the simulation parameters and the redshift influence the evolution of the brightness temperature distribution, we condition the task on those parameters and the time step of the first spatial slice.

At late times, the brightness temperature distribution remains zero for most of the lightcones. We trim the lightcones by keeping the first 100 slices of this period and removing the remaining ones. Additionally, the brightness temperature distribution contains outliers with large absolute values. To regularize that distribution, we effectively clamp the distribution by preprocessing the brightness temperature via

$$\delta T'_{\text{BT}} = \text{sgn}(\delta T_{\text{BT}}) \cdot \ln|\delta T_{\text{BT}}| \; . \tag{38}$$

The resulting values are normalized to zero mean and unit variance. The parameter values and the time step of the first slice are min-max normalized as specified in Eq.(32).

We divide the lightcones into a training, validation and test set, dataset consisting of 3800, 540 and 750 lightcones, respectively. During training, we augment every sublightcone with spatial rotations, reflection and translations. The latter is possible because the lightcones have periodic boundary conditions. More details about the augmentation and processing of the sublightcones can be found in the App. B.

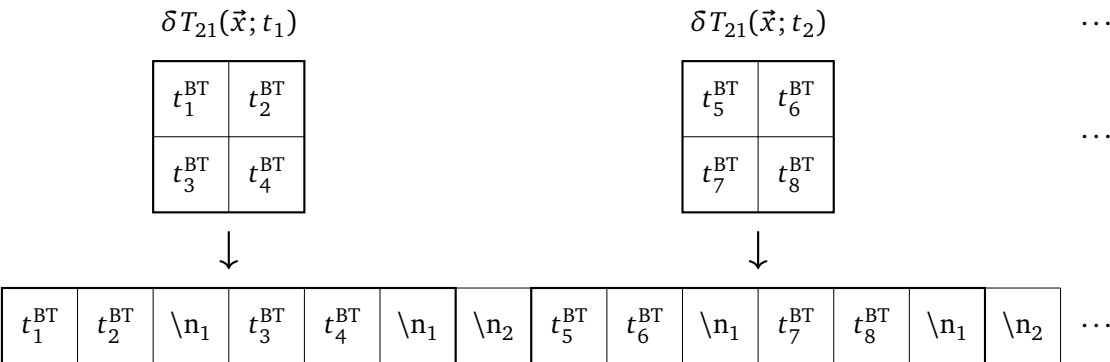

Figure 7: Illustration of the 3D lightcone representation as a sequence of continuous tokens. Every spatial slice, $\delta T_{BT}(\vec{x}; t_i)$, is patched and the resulting grid is flattened. The resulting sequences are concatenated into a single one. Two newline characters, $\backslash n_1$ and $\backslash n_2$, are inserted to keep track of the 3D structure.

**Architecture**   For generation, we introduce 8 input connectors, one for each of the 6 simulation parameters, $\vec{p}$, one for the time step of the first sublightcone slice, $t_{\text{init}}$, and one for the brightness temperature patches, $t^{\text{BT}}$. For the latter, we also add an output connector. Altogether, we want to train a network that encodes the conditional probability, where the condition is represented internally as $z$,

$$p_\theta\left(t_i^{\text{BT}}|z\right) \approx p\left(t_i^{\text{BT}}|t_{i-1}^{\text{BT}}, \ldots, t_1^{\text{BT}}, \vec{p}, t_{\text{init}}\right). \tag{39}$$

To focus again on the backbone, all connectors consist of one affine layer.

To translate the parameterization $z$ into the probability density of Eq.(39) we need a flexible family of distributions which can encode the high inter-pixel correlations of a patch. We use Conditional Flow Matching (CFM) [58, 63, 105], which is known for this ability. It requires another network to learn a velocity field $v_\theta$ that transports samples of a Gaussian distribution to samples of another distribution along an internal time direction, $\tau \in [0, 1]$. By making this vector field dependent on the parameterization $z$, the CFM setup is able to yield the desired mapping

$$\frac{dx(\tau)}{d\tau} = v(x, \tau, z) \quad \text{with} \quad x(\tau) \sim \begin{cases} \mathcal{N}(\mu = 0, \sigma = 1) & \tau = 0 \\ p(t_i^{\text{BT}}|t_{i-1}^{\text{BT}}, \ldots, t_1^{\text{BT}}, \vec{p}, t_{\text{init}}) & \tau = 1 \end{cases}. \tag{40}$$

The network is then trained to regress the vector field interpolating linearly between sampled endpoints,

$$\mathcal{L} = \left\langle (v - v_\theta(x_\tau, \tau, z))^2 \right\rangle_{p(t_i^{\text{BT}}|t_{i-1}^{\text{BT}}, \ldots, t_1^{\text{BT}}, \vec{p}, t_{\text{init}}), \mathcal{N}(x_0; 0, 1), U(\tau; 0, 1)}$$
$$\text{with} \quad v \equiv t_i^{\text{BT}} - x_0 \quad \text{and} \quad x_\tau \equiv (1 - \tau)x_0 + \tau t_i^{\text{BT}}. \tag{41}$$

The network encoding the velocity is a convolutional neural network, which inputs $14 \times 14$ patches with 4 channels. The first two channels contain the transported sample $x_\tau$ and the internal time $\tau$, the latter being expanded to match the patch shape. The output connector of the brightness temperature patches yields a parameterization $z \in \mathbb{R}^{d_z}$ with dimension $d_z = 2 \cdot 14 \cdot 14$, which gets reshaped to fill the remaining 2 channels. The convolutional network consists of a convolutional input layer mapping the 4 channels to 64, 6 residual convolutional layers and an output convolutional layer mapping the 64 channels to 1. A residual layer consists of one convolutional layer mapping the 64 channels to 128 and another convolutional layer with a kernel size of 1 mapping the 128 channels to 64, followed by the residual connection. Every convolutional layer has a kernel size of 3 if not stated differently. The convolutional layers are interleaved with silu activations, and the padding is chosen to keep the patch shape. In total, the convolutional velocity network contains 490k parameters.

The CFM network is the parameterization $\mathcal{P}$ of the conditional distribution in Fig. 2. To obtain this distribution, the ordinary differential equation (40) must be integrated from $\tau = 0$ to $\tau = 1$.

**Training**   Starting from either the Qwen2.5 or random backbone, we finetune L3M

1. completely (360M parameters);
2. with LoRA of rank $r = 8$ (5.5M parameters);
3. with LoRA of rank $r = 2$ (2.2M paramerers); or
4. with frozen backbone (1.0M parameters).

LoRA is applied to all linear layers of the backbone. Additionally, all layer norm weights are trained in the first 3 cases. As for the regression task, the scaling factors of the final layer norm are reinitialized to ones for the pretrained LLM setup. The number of trainable parameters includes the convolutional velocity network. For the L3M setups, we use the prompt

```
<|im_start|>system
<|system-prompt-token|><|im_end|>
<|im_start|>user
<|parameter_0|>...<|parameter_5|><|time_step|><|im_end|>
<|im_start|>assistant
<|lightcone-token|>t₁ᴮᵀ...t₁₀₀ᴮᵀ<|nl_1|><|nl_2|>t₁₀₁ᴮᵀ...<|im_end|>
```

In addition, we also train reference networks from scratch which have a similar number of trainable parameters as the pretrained L3M setups. The autoregressive task forces the reference networks to have a causal attention mask. For these networks, we use the prompt

```
<|system-prompt-token|> <|parameter_0|>...<|parameter_5|>
<|time_step|> <|lightcone-token|> t₁ᴮᵀ...t₁₀₀ᴮᵀ <|nl_1|> <|nl_2|>
t₁₀₁ᴮᵀ...
```

Here, `<|parameter_0|>`, ..., `<|parameter_5|>` and `<|t_index|>` are placeholders for the parameter- and time-step-modalities. Moreover, `<|system-prompt-token|>` and `<|lightcone-token|>` are additional trainable tokens.

We do not extensively optimize the hyperparameters of the reference networks since we focus on the comparison between pretrained and random backbone weights. The selected hyperparameters for the L3M and reference networks are listed in Tab. 3. Since training the randomly initialized L3M backbone completely matches our definition of a reference network, we do not train an additional reference network for this case. The only conceptual difference to the other two reference networks is the chosen prompt template. However, we do not expect any significant difference from replacing it.

## 5.2 Results

We again start by showing the validation loss of different L3M setups in Fig. 8. Each curve is the mean value from five runs and the band covers one standard deviation. The L3M setups differ both by the initialization of the backbone and by the number of finetuned parameters, and we group the networks by the latter in Fig. 8 in descending order from left to right. Due to the data augmentation, every instance of a batch can be viewed as a new sample. We therefore show the loss as a function of the batch number.

| | |
|---|---|
| Batch size | 64 |
| Epochs | 20 |
| Learning rate | $5 \cdot 10^{-5}$ |
| Learning rate schedule | 1 epoch linear warmup, 18 epochs stable, 2 epochs cosine decay |
| Weight decay | $10^{-3}$ |
| Max. gradient norm | 1 |
| Attention dropout | $10^{-2}$ |
| Optimizer | AdamW |

| | | | |
|---|---|---|---|
| Hidden dim | 128 | 128 | 256 |
| Transformer blocks | 4 | 7 | 7 |
| Query heads | 4 | 4 | 4 |
| Key-Value heads | 4 | 4 | 4 |
| MLP hidden dim | 128 | 424 | 560 |
| Number of parameters | 1.03M | 2.18M | 5.48M |

Table 3: Training (left) and reference network (right) hyperparameters. The number of parameters match the number of trainable parameters L3M networks with partially finetuned backbone.

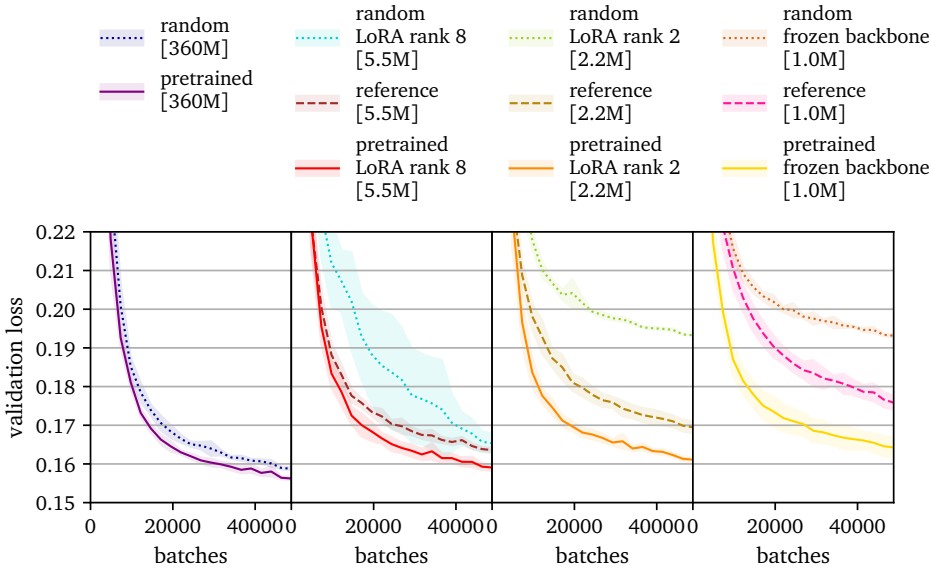

Figure 8: Mean validation loss with a $1\sigma$-band determined from 5 runs.

Similarly to the regression task, we find that the best performing setup at each number of trainable parameters is the pretrained LLM backbone. This holds for the case with maximal number of trainable parameters, where the random backbone is essentially a dedicated network. When using LoRA finetuning or completely freezing the backbone, only the L3M with pretrained backbone networks improve on the dedicated network. In particular at LoRA rank 2, the L3M with random backbone essentially fails as we demonstrate later on. This is expected to some degree, since it is difficult to introduce meaningful structure to the backbone through rank 2 modifications. Meanwhile, the LLM backbone can be effectively finetuned beyond the performance of the dedicated network, even at LoRA rank 2. These results therefore highlight the advantage offered by pretrained backbones, where the existing structure can be repurposed for the task at hand even through finetuning only a small fraction of the weights. This observation is supported by freezing the backbone completely. The loss of the pretrained and the random L3M degrade only marginally compared to LoRA with rank 2. On the one hand, this implies that LoRA of rank 2 modifies the backbone weights very mildly. On the other hand, the pretrained LLM backbone is able to generate lightcones without any finetuning and as such, its language capabilities remain unchanged.

The loss in Fig. 8 measures how well the network is able to regress the CFM target vector field specified in Eq.(41). To determine how well the network approximates the conditional distribution from Eq.(39), we compute the MSE for individual predicted patches in the test set. First, we load the checkpoint with the best validation loss for each run. Then, we sample the next patch conditioned on the preceding ground truth patches, and finally compute the MSE relative to the ground truth. The resulting MSE together with the variation between different runs are stated in Tab. 4. The order of network performance remains the same. However, this measure amplifies the difference between the pretrained and randomly initialized backbone finetuned with LoRA of rank 2 and with frozen backbone.

As a validation that the networks generate coherent lightcone slices, we present samples in Figure 9. These samples belong to the same representative lightcone where for different time steps two consecutive slices are autoregressively generated with a context of 10 preceding slices. First, we compare the coherence of each slice. For the pretrained and completely finetuned backbone, the grid structure of the patching is almost invisible. While this is marginally

| L3M setup | trainable parameters | next-patch MSE | MSE ratio to pretrained |
|---|---|---|---|
| pretrained | 360M | $0.08393 \pm 0.00021$ | - |
| random | 360M | $0.0871 \pm 0.0003$ | $1.038 \pm 0.005$ |
| pretrained LoRA rank 8 | 5.5M | $0.0875 \pm 0.0012$ | - |
| reference | 5.5M | $0.0951 \pm 0.0003$ | $1.088 \pm 0.015$ |
| random LoRA rank 8 | 5.5M | $0.098 \pm 0.004$ | $1.12 \pm 0.05$ |
| pretrained LoRA rank 2 | 2.2M | $0.0930 \pm 0.0017$ | - |
| reference | 2.2M | $0.1039 \pm 0.0028$ | $1.12 \pm 0.04$ |
| random LoRA rank 2 | 2.2M | $0.2232 \pm 0.0008$ | $2.40 \pm 0.04$ |
| pretrained frozen backbone | 1.0M | $0.099 \pm 0.009$ | - |
| reference | 1.0M | $0.116 \pm 0.005$ | $1.18 \pm 0.11$ |
| random frozen backbone | 1.0M | $0.2186 \pm 0.0004$ | $2.22 \pm 0.19$ |

Table 4: MSE for next patch prediction, as described in the text. The different L3M setups are grouped in the same way as in Fig. 8. The uncertainties are the standard deviations of the different runs.

worse for the pretrained L3M network finetuned with LoRA of rank 2, the randomly initialized network finetuned with LoRA of rank 2 sometimes fails to generate coherent slices. This is most apparent in the middle rows of Figure 9. Next, we compare the large scale structure of the lightcone slices. For both pretrained backbones, the generated large scale structure agrees with the ground truth. However, this is not the case for the random backbone, which is most dominant in the lower rows of Figure 9. We conclude that both pretrained backbone setups are able to generate coherent lightcone slices, while the randomly initialized network finetuned with LoRA of rank 2 is only able to generate typical patches for the corresponding brightness temperature value range. In particular, it is unable to forecast the evolution of the large scale structure. Generated slices of the L3M setups with frozen backbones are shown in App. C, revealing that the pretrained L3M is still able to generate coherent lightcone slices.

# 6   Conclusion

The impressive success of pretrained transformers in fundamental physics begs the question of whether the largest and most general pretrained networks, LLMs, can be used for physics data. In this paper, we have shown for the first time that LLM backbones can be successfully finetuned for SKA data. We introduced a scheme for adapting an LLM to new modalities via connector networks and applied it to the 0.5B Qwen2.5 language model. To judge the specific advantage offered by the pretrained weights, we compared to a baseline with the same LLM architecture but with completely re-initialized weights.

As benchmark tasks for our Large Lightcone Language Model (L3M), we studied (i) regression of cosmological and astrophysical parameters from the spatially-averaged 21cm signal and (ii) forecasting full 21cm lightcone slices. In both tasks, we found that pretrained LLM weights improve the performance over the re-initialized baseline for a given number of training iterations, indicating improved convergence and data-efficiency. Interestingly, the advantage can be amplified by wrapping the input data with a chat-style template.

We also compared L3M to dedicated reference networks. The pretrained LLM backbones outperformed references with a matching number of trainable parameters. This is also true for the randomly-initialized backbone in the regression task, and we identified a performance

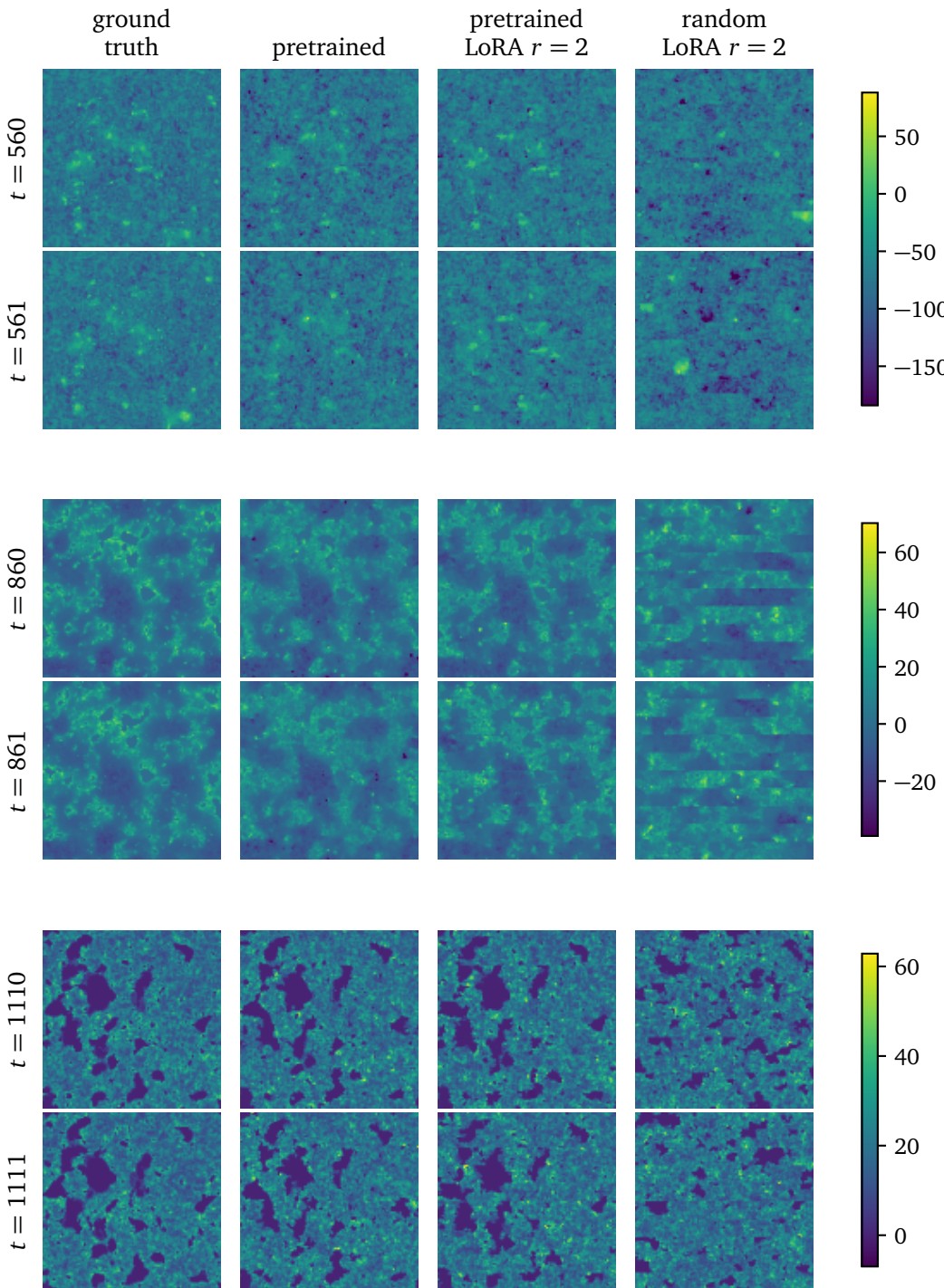

Figure 9: Forecast slices as described in the text for different time steps $t$ by a pretrained L3M setup which is either completely finetuned or partially finetuned with LoRA of rank 2 and a randomly initialized L3M setup finetuned with the same LoRA configuration. The pixel values are the post-processed brightness temperatures.

scaling with the number of transformer blocks in the network. For the more difficult generative task, the random backbones did not improve on a dedicated network. However, pretrained and finetuned LLM backbones retain their advantage. Our results suggest that pretrained weights from LLMs offer a strong initialization, even for truly out-of-domain fundamental physics tasks. Whether our observed efficiency and performance gains justify the use of LLMs in fundamental physics needs to be evaluated for a given specific application.

## Code availability

The code for this project is published at https://github.com/heidelberg-hepml/L3M.

## Acknowledgements

We thank Nina Elmer, Henning Bahl and Ramon Winterhalder for showing us how to unbias the heteroskedastic covariance matrix.

CH's work is funded by the Volkswagen Foundation. This work is supported through the KISS consortium (05D2022) funded by the German Federal Ministry of Education and Research BMBF in the ErUM-Data action plan, by the Deutsche Forschungsgemeinschaft (DFG, German Research Foundation) under grant 396021762 – TRR 257: *Particle Physics Phenomenology after the Higgs Discovery*, and through Germany's Excellence Strategy EXC 2181/1 – 390900948 (the *Heidelberg STRUCTURES Excellence Cluster*). We acknowledge support by the state of Baden-Württemberg through bwHPC and the German Research Foundation (DFG) through grant INST 35/1597-1 FUGG.

| | |
|---|---|
| Hidden dim | 896 |
| Transformer blocks | 24 |
| Query heads | 14 |
| Key-Value heads | 2 |
| MLP hidden dim | 4,864 |
| MLP activation | silu |
| RMS norm $\epsilon$ | $10^{-6}$ |
| RoPE base frequency | $10^6$ |
| Shared embedding weights | ✓ |
| Parameters | 0.49B |
| Non-embedding parameters | 0.36B |
| Max sequence length | 32,768 |
| Max generation length | 8,192 |
| Vocabulary size | 151,936 |

Table 5: Network hyperparameters of Qwen2.5-0.5B.

## A  Specifics about Qwen2.5

In this section, we review the hyperparameters of the Qwen2.5-0.5B-Instruct LLM and the general Qwen2.5 training setup [50]. The network hyperparameters are stated in Tab. 5 and the training hyperparameters in Tab. 6.

The pretraining has been split into two stages: First, the networks are trained with sequences consisting of 4,096 tokens and a RoPE base frequency of $10^5$. Then, this base frequency is enlarged to $10^6$ and the network is trained on sequences with $32,768$ tokens.

After that, the Qwen2.5 networks undergo a supervised finetuning stage, with focus on generating long sequences with $8,192$ tokens, following instructions, understanding structured data, and reasoning.

Finally, the networks are trained with Reinforcement Learning in two stages. In the first stage, the networks are trained with Direct Preference Optimization, which only requires one positive and one example for every query. This labeling has been created semi-automatically, without the use of any reward model. In the second stage, a reward model is trained. Another dataset is joined with the previous Reinforcement Learning dataset. In this stage, Group Relative Policy Optimization is used, sampling 8 responses per query.

## B  Generation dataset

In advance of training, the lightcones are split into sublightcones consisting of 5 initial slices and 45 remaining ones. To split a lightcone, we iterate along the time direction with a step size of 45. This guarantees that the initial 5 slices (except the very first ones) are also part of another sublightcone for which the next-patch loss can be computed. During the sampling of a batch, a random interval of 12 slices is chosen. In this way, the network is able to see different initial slice configurations.

To reduce the RAM footprint, each lightcone slice is loaded in FP16. After sampling a batch, the lightcones are converted to FP32 and afterwards, the preprocessing defined in Eq.(38) is applied.

| Pretraining: | |
|---|---|
| Dataset size | 17T tokens |
| Sequence lengths | 4,096 - 32,768 |
| **Supervised finetuning:** | |
| Dataset size | $\sim$ 1M examples |
| Sequence length | 32,768 |
| Epochs | 2 |
| Learning rate | Decay from $7 \cdot 10^{-6}$ to $7 \cdot 10^{-7}$ |
| Weight decay | 0.1 |
| Gradient clipping | 1.0 |
| **Reinforcement Learning I:** | |
| Dataset size | $\approx$ 150,000 examples |
| Epochs | 1 |
| Learning rate | $10^{-7}$ |
| **Reinforcement Learning II:** | |
| Batch size | 2048 |

Table 6: Training hyperparameters for Qwen2.5 models.

## C  Generation with frozen backbone

In this section, we show generated lightcone slices of the pretrained and random L3M setups with frozen backbones in Fig. 10. The slices have been generated analogously to the ones of section 5.2 and the results are very similar. The pretrained L3M with frozen backbone generates marginally worse slices, the random L3M with frozen backbone fails.

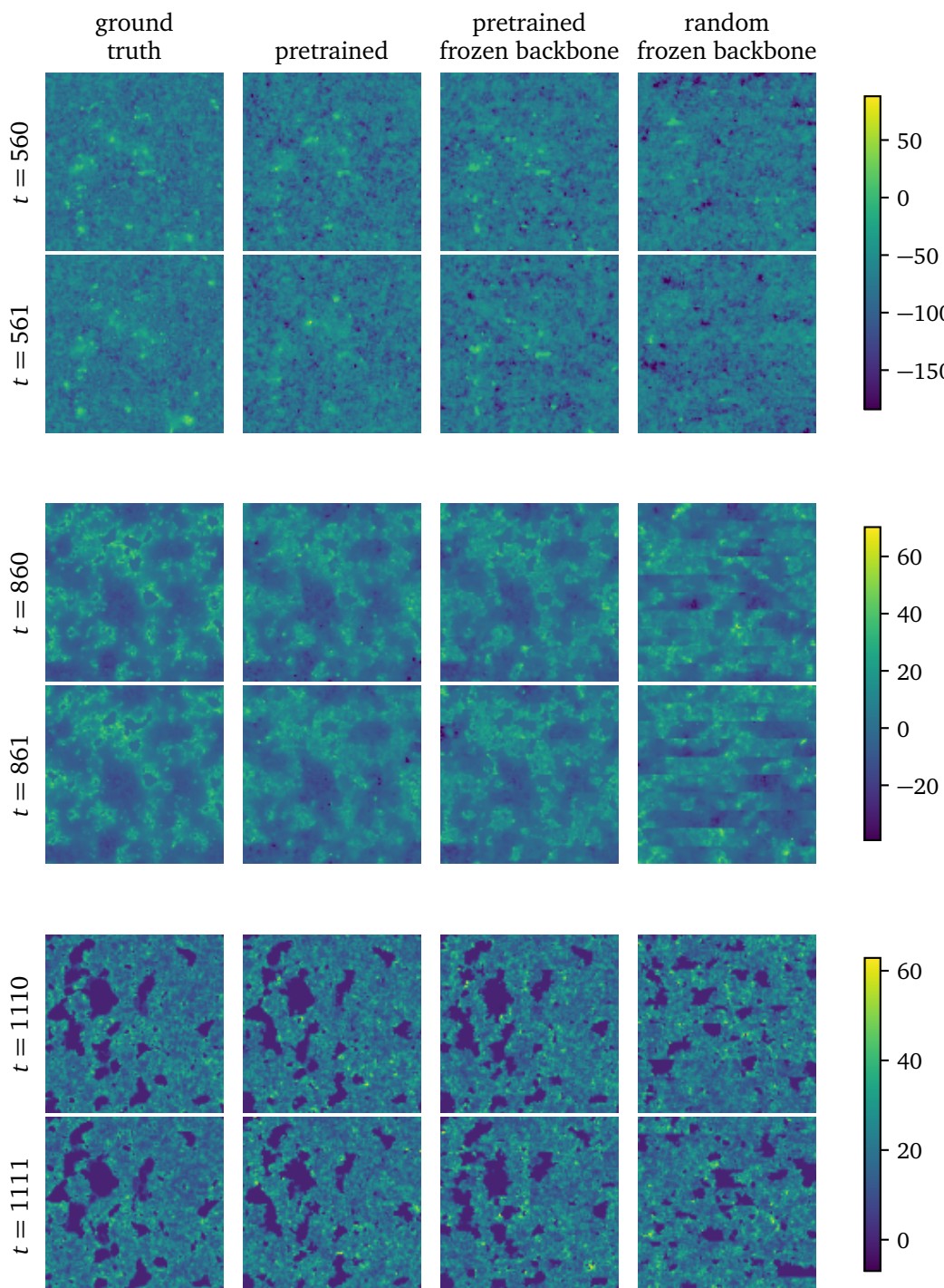

Figure 10: Forecast slices as described in the text of section 5.2 for different time steps $t$ by a pretrained L3M setup with a backbone that is either completely finetuned or completely frozen and a randomly initialized L3M setup with a completely frozen backbone. The pixel values are the post-processed brightness temperatures.

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
