# Peer review of "Large Language Models -- the Future of Fundamental Physics?"

_SciPost Physics_

## Round 1 · Referee Report · Anonymous (Referee 2) · 2025-10-1

Report

This paper pioneers work to explore if LLMs and their structural learned representation can be exploited for physics application. This question is explored for the first time quantitatively and in detail.

The idea is to re-purpose an LLM backbone for physics data in analogy to finetuning. The modality is changed from language to physics data, a sort of model reprogramming.

Data of 21cm background fluctuations and associated standard cosmological tasks are used as a testbed for this re-programmed LLM.

The first test is to completely freeze the backbone transformer, training only the connectors, and compare against
a network where the weights of the backbone transformer are re-initialized.

Comparison to a small network illustrating the performance for a comparable number of trainable parameters
as the L3M connectors and a large network illustrating the ultimate performance of a dedicated network are provided.

Improvements are seen for the pre-trained vs. the randomly initialized network. This is demonstrated as a lower loss in Fig. 4. However, the difference between random and pre-trained is comparatively small wrt the two references [32k and 990k]. This indicates that there is still a lot of room for improvement [to the 990k] and that the 32k model is too small to get a good enough performance (also consolidated by Fig. 5).

Interestingly, Fig. 6 does not show an appreciable difference between any of the three: pre-trained, random, 990k. The reader is wondering if there is not a task and associated metric which has a higher sensitivity to the variation of models?

The second set of results allows for a fine-tuning of the backbone. Also, the downstream task changes.

The reader would, however, have liked to see the same task with and without fine-tuning.

Fig. 8 shows a similar trend as Fig. 4.

LoRA is presented as a promising technique to safe compute and finetune only a small fraction of the weights.

Requests:

Fig. 8: please align legend to the three sub-figures for easier interpretation of the results.

Table 4: difficult to compare numbers which are mainly similar in the first 2-3 digits. The reader is wondering if ratios could be shown instead or some other visual representation could be found.

Visual inspection of Fig. 9 is very difficult. It would be useful to have an associated performance metric to guide the reader.

Summary:

Overall, the paper is very original and certainly explores new ground here. The results are promising, though it remains to be seen if the marginal improvements lead to real measurable physics gains in the matrix of compute resources vs. performance.

Recommendation

Ask for minor revision

  • validity: high
  • significance: good
  • originality: top
  • clarity: high
  • formatting: excellent
  • grammar: excellent

Author:  Daniel Schiller  on 2025-12-12  [id 6140]

(in reply to Report 2 on 2025-10-01)

We thank the referee for their comments, which we address in turn below: 1. The large gap is due to the large reference network yielding a smaller spread of the regressed values, especially in $\Omega_m$ and at low values of $m_{\text{WDM}}$. We added this clarification to the text. Since this is visible in Fig. 6, we do not include any additional metric. 2. We thank the referee for this recommendation. We included those experiments with the result that the pretrained L3M is able to generate lightcone slices with a completely frozen backbone. We also added example generated slices to appendix C. 3. We aligned the legend which makes the interpretation easier. 4. We included the ratio to the MSE of the pretrained L3M analog. 5. We agree that visual inspection of Fig. 9 is difficult at first sight. However, we believe that using a quantitative metric to compare individual lightcone slices could be misleading since the brightness temperature evolution is not deterministic. In Tab. 4 we show an error metric and we believe it is meaningful because we compute it over many lightcones. The goal of Fig. 9 is rather to show qualitatively that the generated results are ‘physically’ similar to the ground truth. The main observation is that the random backbone yields clear artifacts. We added text to guide the reader: First, we focus on the coherence of each image. Then, we inspect the large-scale structure. We hope this makes the figure easier to ingest.

---

## Round 1 · Referee Report · Anonymous (Referee 1) · 2025-10-1

Strengths

The paper presents a study on whether pretraining on out-of-domain data (i.e. language) can be useful for analyzing numerical data from the Square Kilometer Array, if the scale of the pretraining is large enough. This type of study has not, to my knowledge, been done before in this particular field.

Weaknesses

The paper could be more clear when it comes to the exact setup used for their studies (see comments/requested changes).

Report

The paper meets the first two acceptance criteria (Expectations) of the journal (Provide a novel and synergetic link between different research areas; open a new pathway in an existing or a new research direction, with clear potential for multi-pronged follow-up work). It also in principle meets the general acceptance criteria, but I do ask for some clarifications as outlined below. With these remarks addressed, I would recommend this paper for publication in SciPost Physics.

Requested changes

1. Introduction: The claim “transformers have been shown to be the best-suited architecture for both representation learning and generation” lacks a reference.
2. Introduction: recommend to also cite JetClass II in the [40,41] citation bracket.
3. Introduction: The authors ask the question whether the extreme gap in scale between LLMs and the typical physics networks can compensate for the data modality shift. I would like a bit more motivation behind the idea, for example a few references (if I understand ref 44 correctly, this is not related directly to this specific question as they don’t use the LLM directly on numerical data), as one normally wouldn’t think there would be anything in the structure of language that would be applicable to physics data, given that the modalities are so wildly different.
4. Section 2: In order for this section to be useful to a reader that is not already familiar with all of the material, care needs to be taken to explain all elements introduced in the expressions and equations. Simple example: the theta in eq 3. It’s obvious to me what this theta refers to, but this might not be the case for the general audience (especially if this audience is expected to need all this detail to understand how the model works). Another example is eq 22 where the policy is introduced, but it’s not mentioned what policy means or what the significance of it is.
5. Section 2: Consider cutting or moving items that are not crucial for the understanding of the paper, in order to avoid confusing the reader. For example, the concept of RL does not appear in the main text, only the appendix.
6. Section 3: Towards the end of section 3.1, it would be good to either 1) mention what type of data you expect to input as the linguistic-coded token, as this is quite mysterious at this point, or 2) refer to the section in which this will be explained.
7. Section 3: Beginning of section 3.2: does SKA have a reference? I also believe the acronym is not spelled out anywhere in the paper, apart from inside a prompt box.
8. Section 3: A few selected visualizations of the data would be helpful here.
9. Section 4: Heteroskedastic loss does not belong to the most common loss functions. A reference here would be useful.
10. Section 4: Eq 36, motivate this choice or mention what it is called.
11. Section 4: Add a line explaining how eq 37 illustrates the point you want to make about unbiasing the covariance matrix.
12. Section 4: It is at this point still unclear to me how the “mixed” setup of section 3.1 (linguistic token and numerical token) works. This has to be explained at the latest at this point, as the text mentions tokens that the LLM already has pretrained embeddings for.
13. Section 4: It is unclear whether the idea to update the weights of the connector blocks separately is the authors’ own invention. If not, a reference or statement of the origin should be added. And similarly for the idea to duplicate the test dataset, original invention or reference/origin should be stated.
14. Section 4: Re-initialization of the scaling factors of the final layer norm – is this an empirical find, or something known in the literature (in which case a reference is needed)?
15. Section 4: Why do the two reference networks not have a causal attention mask? What motivates introducing this difference to the original Qwen architecture?
16. Section 4: The larger of the reference networks is said to “illustrate the ultimate performance of a dedicated network”, but it is then stated that they authors “do not care about its best possible performance” in the context of hyperparameter selection. The statements appear contradictory. Was any type of hyperparameter scan performed at all, in order to roughly get the “ultimate” performance out of this network?
17. Section 4: In general, the reasoning behind the choice of reference networks should be made clearer, with any caveats explicitly discussed. Qwen is several orders of magnitude larger than the reference networks, but must on the other hand adjust to the new domain.
18. Section 4: Fig 4, top row, center and right plots are missing one gridline.
19. Section 4: The observation in Fig 5, that increasing the depth of a randomly initialized backbone helps the performance, even though the backbone is not trained, is very interesting. I would like to have seen a discussion about this phenomenon, including relevant references.
20. Section 4: The method used to produce fig 6 is not entirely clear, please rephrase to make it easier to understand.
21. Section 5: The sequence length is kept reasonable by restricting to 12 consecutive spatial slices. How many slices do these lightcones normally have?
22. Section 5: It is not entirely clear to me where the CFM is plugged in, or where LoRa is applied. Perhaps a sketch of the complete architecture at this point would help the reader understand how the different parts fit together. This would also make it clearer what is actually being finetuned, as described on p 20.
23. Section 5: It seems the reference networks in this section are chosen in a different way compared to in section 4. Please explain the reasoning behind how the reference networks were chosen here.
24. Section 5: Fig 8 caption is confusing -- do all curves show the average of 5 finetunings, except 1 curve that shows the average of 7?
25. Section 6: “the pretrained LLM backbone outperforms the reference networks” – is this true also for the parameter estimation case? The end of section 4 seemed to suggest that the large reference network performed better.
26. Section 6: “For the regression, we also observe this behavior in the randomly-initialized network and identified a scaling with the number of transformer blocks in the network.” – I don’t see this mentioned in the main text.
27. References: Check capitalization in titles

Recommendation

Ask for minor revision

  • validity: top
  • significance: top
  • originality: top
  • clarity: high
  • formatting: excellent
  • grammar: perfect

Author:  Daniel Schiller  on 2025-12-13  [id 6146]

(in reply to Report 1 on 2025-10-01)

We thank the referee for their comments, which we address in turn below:

  1. We agree that that statement is strong enough to warrant a reference, but we opted to simply remove it, since we found that it did not meaningfully contribute to the message of the paragraph.

  2. We have included the citation.

  3. We restructured this paragraph to try to clarify the motivation, which begins two paragraphs earlier: modern applications of machine learning already explore the trade-off between physical biases (equivariance) versus scale (self-supervised pretraining) and this paper extends this idea to the extreme case. We removed the phrasing that this is an “obvious” question, which may have discounted the preceding argument. As far as we are aware, this is the first study of its kind and we cannot point to similar references.

  4. We defined theta and we slightly reworded the text around Eq. 22 to make it clearer that the policy is the LLM.

  5. We pointed out that we include section 2.4 for completeness as finetuning is an important part for obtaining the LLM weights.

  6. We referred to the corresponding sections where we define the numerical tokens.

  7. We expanded the first use of the acronym and added references in the introduction. After uploading the resubmission, we became aware of the small artefact in the beginning of section 3.2, which we have already fixed in our local version.

  8. We referred to sections 4 and 5 where visualizations can be found.

  9. We included a reference.

  10. We mentioned the name Cholesky decomposition and motivated its application.

  11. We expanded the explanation of the example (37).

  12. First, we mentioned that the prompts are visualizations of the input sequences. Then, we added a sentence stating that numerical tokens are processed with the input connectors C and that the linguistic tokens are processed with the embedding map E.

  13. We did not take this technique from any paper, but at the same time cannot claim that it is an original invention of ours. For this reason, we opted to simply add a sentence explaining that this was our empirical finding.

  14. This was our empirical finding. We included this in the text.

  15. Removing the causal attention mask gives the reference networks the advantage that every token gets attended to every other. We added this explanation in the text.

  16. We agree that the original wording was contradictory. We updated the description of the large reference network to “Illustrating the expected performance of an unrestricted and dedicated network”. The importance of including the large reference is simply to benchmark the conventional approach of throwing a medium-sized network at this problem. Since it already outperforms the LLM setups, we do not see a benefit to optimizing the hyperparameters of this network.

  17. We reworded the description of both reference networks to focus on the motivation. The small reference network matches the number of trainable parameters of the L3M setup. The large network has many more trainable parameters as the L3M setup. Its motivation is explained in our response to the referee’s previous comment above.

  18. We fixed this.

  19. We included a small supposition for this phenomenon. However, we have not studied this in detail.

  20. We expanded the description of this procedure.

  21. We mentioned that one lightcone consists of 2350 spatial slices.

  22. Figure 2 is the most general L3M setup. To avoid introducing duplicate figures, we referred to Fig. 2 and mentioned that the CFM is the parameterization P. Further, we mentioned that the velocity field must be integrated to obtain the conditional probability distribution of the patches. Also, we stated that LoRA is applied to all linear layers of the backbone.

  23. The motivation is the same as for the small reference network of the regression task: We match the number of trainable parameters. Unfortunately, the work “trainable” was missing in the text, which we have included. In addition, we stated that the autoregressive task forces the reference networks to have a causal attention mask.

  24. To avoid confusion, we restricted setups to have 5 runs.

  25. We reworded this part to emphasize that the LLM only beats the references with an equal number of trainable parameters.

  26. This refers to our observation that the randomly-initialized backbone improves on the small reference network (Figures 4 and 5). We also reworded this text to improve clarity.

  27. Does the referee have specific capitalization errors in mind? As far as we can tell, our reference titles match those provided by INSPIRE.

---

## Round 1 · Referee Report · Anonymous (Referee 3) · 2025-10-8

Report

This manuscript presents a novel investigation into the application of pretrained Large Language Models (LLMs) for numerical tasks in fundamental physics. The authors adapt the Qwen2.5-0.5B model to analyze and generate simulated cosmological data from the Square Kilometre Array (SKA). The core methodology involves replacing the LLM's standard embedding layers with “connector networks” to interface with numerical physics data, effectively reprogramming the model for a new modality. The authors systematically compared the performance of their adapted LLM against several baselines and showed that the out-of-domain pretraining on linguistic data provides a significant and consistent performance advantage in the context of a small dataset (~500 0 samples in their experiments). Overall, the paper is well-structured and clearly written. I listed below a few questions that should be addressed or clarified before publication:

  • The study is based on the Qwen2.5-0.5B model. In the LLM world, this is a relatively small model. A key question then is how the results shown in this paper scale with the model size. Would a 7B, 70B, or even larger model provide a correspondingly larger advantage or not? While running experiments with larger models is likely out of scope, some discussion/comments on the scaling behavior would be very helpful.

  • A related question is the scaling behavior with the sample size. An important question is whether the LLM-based approach is universally advantageous for all dataset sizes, or only when the available dataset is small (and if so, where is the turning point). Again, running experiments with more data is likely beyond the scope due to resource limitations, but it would be nice if the authors could comment on this or refer to existing literature in other fields.

  • p.15: In the “Training and reference networks” paragraph, it is mentioned that “In addition, we insert a copy of the test dataset into it.” Could you please clarify how it is done? Does it leak any truth information from the test dataset into the training?

  • p.16, Figure 4: Why are the loss curves of the two reference networks shown as flat lines?

  • Figure 4: From these plots, one would conclude that training the larger reference network from scratch gives much better results than adapting the pretrained LLM. Is that correct? If so, what is the benefit of adapting an LLM?

  • Figure 4: Would finetuning the backbone further improve the performance?

  • p.18, 2nd paragraph: Should it be 12,000 patches, or 1,200 (= (140140) / (1414) * 12)?

  • Table 3: For the finetuning experiment, is the same learning rate applied to all layers, or different LRs are used for the backbone and the connectors?

Recommendation

Ask for minor revision

  • validity: good
  • significance: ok
  • originality: high
  • clarity: high
  • formatting: excellent
  • grammar: excellent

Author:  Daniel Schiller  on 2025-12-12  [id 6141]

(in reply to Report 3 on 2025-10-08)

We thank the referee for their comments, which we address in turn below: 1. We would prefer not to include specific comments in the paper because we can only speculate on the results of such experiments. On the one hand, the larger LLMs encode more structure which may be useful. However, that structure is accompanied with a larger embedding dimension, for which training the connector networks is likely harder. 2. Similarly to the previous question, we do not want to speculate since we do not have results for those points. However, we do not see a reason why the pretrained LLM backbone should stop outperforming the random LLM backbone at any dataset size. At the same time, we believe that for large enough datasets, the reference network should saturate the performance of the pretrained LLM performance. 3. This was a typo. We corrected this to “training dataset” and rephrased the sentence. 4. We have trained the reference networks for many more epochs, which is why we only show the best loss with a horizontal line. We have elaborated this in the text as well. 5. Indeed the large reference is better for this specific regression task, so LLMs do not provide a concrete benefit here. The purpose of Figure 4 is rather to expose the difference to a random initialization and especially the effect of the chat templates. 6. It will improve the performance most likely, especially since it will increase the number of trainable parameters. 7. This was a typo. The correct value is 1,200 which we corrected in the text. 8. Yes, the same learning rate has been applied to all layers.

---

## Round 2 · Referee Report · Anonymous (Referee 3) · 2025-12-18

Report

The authors have addressed most of my comments in the revised manuscript. I believe it is suitable for publication in SciPost Physics.

Recommendation

Publish (meets expectations and criteria for this Journal)

---

## Round 2 · Referee Report · Anonymous (Referee 2) · 2025-12-19

Report

I am happy with the replies.

Recommendation

Publish (meets expectations and criteria for this Journal)

---

## Round 2 · Author Response

We thank all referees for their comments. We revised our paper based on the referees' comments and replied to each referee individually.

---

## Round 2 · List of Changes

We highlighted major changes with an orange font.

---

## Editorial Decision

in_refereeing